# Advanced Technology Evolution Pathways of Nanogenerators: A Novel Framework Based on Multi-Source Data and Knowledge Graph

**DOI:** 10.3390/nano12050838

**Published:** 2022-03-02

**Authors:** Yufei Liu, Guan Wang, Yuan Zhou, Yuhan Liu

**Affiliations:** 1Center for Strategic Studies, Chinese Academy of Engineering, Beijing 100088, China; liuyf@cae.cn; 2National Numerical Control Systems Engineering Research Center, School of Mechanical Science and Engineering, Huazhong University of Science and Technology, Wuhan 430074, China; m202077010@hust.edu.cn (G.W.); liuyh_0116@hust.edu.cn (Y.L.); 3School of Public Policy and Management, Tsinghua University, Beijing 100084, China

**Keywords:** nanogenerator, technology evolution pathway, knowledge graph, representation learning, multi-source data

## Abstract

As an emerging nano energy technology, nanogenerators have been developed rapidly, which makes it crucial to analyze the evolutionary pathways of advanced technology in this field to help estimate the development trend and direction. However, some limitations existed in previous studies. On the one hand, previous studies generally made use of the explicit correlation of data such as citation and cooperation between patents and papers, which ignored the rich semantic information contained in them. On the other hand, the progressive evolutionary process from scientific grants to academic papers and then to patents was not considered. Therefore, this paper proposes a novel framework based on a separated three-layer knowledge graph with several time slices using grant data, paper data, and patent data. Firstly, by the representation learning method and clustering algorithm, several clusters representing specific technologies in different layers and different time slices can be obtained. Then, by calculating the similarity between clusters of different layers, the evolutionary pathways of advanced technology from grants to papers and then to patents is drawn. Finally, this paper monitors the pathways of some developed technologies, which evolve from grants to papers and then to patents, and finds some emerging technologies under research.

## 1. Introduction

As a novel energy solution for micro and wearable wireless electronic devices, nanogenerators (NG) have been developed to harvest energy from the environment, including biomechanical energy, solar and wind energy, thermal energy, etc. [1]. Based on different physical effects, nanogenerators can be roughly divided into piezoelectric nanogenerators (PENGs), triboelectric nanogenerators (TENGs), and pyroelectric nanogenerators (PYENGs) [2]. Notably, nanogenerators present widespread applications other than energy harvesting, benefiting from related technologies such as 5G and Internet of Things (IoT) [3], nanomaterials [4], flexible sensors [5], and so on. To date, these applications can be divided into two domains. One is the innovative devices and techniques in the engineering domain (e.g., self-powered sensing systems, wearable devices [6]), the other is the biomedical domain (e.g., implantable devices, tissue regeneration [7]). Due to the rapid development and diversity of nanogenerator technology, identifying and understanding the evolutionary path of nanogenerator technology is crucial for decision-makers to capture the development trends and directions [8].

Some previous studies roughly described a sub-field development path of nanogenerator technology based on literature reviews. However, with the rapid development of nanogenerators, the corresponding increase in literature makes it difficult to thoroughly analyze the evolutionary trends of the nanogenerator technologies based solely on literature reviews. Therefore, quantitative approaches such as bibliometrics, patent citation analysis, technology roadmap, and text mining are used to analyze the evolutionary trends [9,10]. However, citation networks only make use of the explicit correlation of data, which ignores the rich semantic information contained in them. To deal with this, the knowledge graph (KG), a constructed knowledge base with powerful semantic processing ability, is taken into consideration naturally [8]. Essentially, the knowledge graph is a semantic network with nodes and edges that reveals the entities and relationships and can formally describe things and relationships in the real world. In addition, at the level of research purpose, scholars only focused on the technology evolution pathways over time [11,12,13,14,15,16,17,18,19], which ignored the progressive evolutionary process from scientific grants to academic papers and then to patents, while technologies generally emerge with grants or papers and become sophisticated with patents.

In this paper, we propose a novel framework for monitoring the evolutionary paths of nanogenerator technology based on analyzing grants, papers, and patents data. The framework is shown in Figure 1. After multi-source data acquisition, the knowledge graph was constructed to capture semantic information between entities, as shown in the top right corner of Figure 1. Different colors of dots show the different types of entities (such as the author, paper, institution, and journal in paper knowledge graph), and the connections between dots show the relations between entities. Then, representation learning and clustering methods were used to cluster entities with similar topics, as shown in the bottom right corner in Figure 1, while the circles represent clusters and the black dots represent the grants, papers, and patents contained in clusters. Finally, we describe the evolutionary path from grants to papers and then to patents by connecting similar clusters.

The key contributions of this paper can be summarized as follows:The nanogenerator field is emergent and rapidly developed, making it hard to analyze the evolutionary pathways of advanced technologies. This paper proposed a novel framework to monitor the evolution pathways based on multi-source data and a knowledge graph.When monitoring the evolution pathways, we applied the representation learning method and clustering method to connect similar entities, which enables the quantitative analysis of large-scale data, thus improving efficiency and accuracy.This paper used multi-source data from three data sources and analyzed the evolutionary pathways between different data sources, which reflected the technology trends comprehensively and pluralistically.

## 2. Literature Review

### 2.1. Development of Nanogenerators

With the rise of the Internet of Things (IoT), advanced materials, and electronics, wearable and implantable devices have developed rapidly. Miniaturization and power continuity have become an important development direction of such devices, which puts high demands on power supply systems [20]. Traditional power methods such as lithium batteries and lead-acid batteries have the limitations of considerable size, short service life, poor flexibility, the possibility of environmental pollution, and the need for frequent replacement. Therefore, developing a new microelectronic power supply device with high flexibility and a sustainable power supply has become the focus of researchers.

Piezoelectric nanogenerators (PENG) using ZnO nanowires were first invented in 2006 by Wang Zhonglin based on the piezoelectric effect to harvest mechanical energy and convert it to electric power, which marked the beginning of self-power technology [21]. After that, other researchers made many attempts and improvements in piezoelectric materials. At present, the mainstream and mature piezoelectric materials include ZnO, BaTiO_3_ [22], lead zirconate titanate (PZT) [23], and polyvinylidene fluoride (PVDF) [24]. While developing piezoelectric materials, triboelectric nanogenerators (TENG) came out in 2012, which is based on the conjunction of triboelectrification and electrostatic induction [25]. Compared with PENG, TENG has the advantages of having a high output, low cost, simple structure design, and excellent stability. Up to now, PENG and TENG have made significant progress in output performance, sensitivity, energy conversion rate, flexibility, and being environmentally friendly [26]. At the same time, some other types of nanogenerators have been developed, such as pyroelectricity nanogenerators (PYENG) and piezoelectric triboelectric hybrid nanogenerators (PTENG) [27].

### 2.2. Technology Evolution Pathways

As a law of nature, evolution occurs all the time. Additionally, there is also an evolution process in the field of technology [28]. At present, the definition of technology evolution is not unified. There are roughly two views among researchers: one holds that technology evolution is generated by the accumulation of continuous innovation behind technology, and the other holds that the development and change process of technology itself symbolizes technology evolution and the induction and display of various changes in the form of paths is the technology evolution pathway [29,30].

The early analysis methods of technology evolution were mainly qualitative methods, including morphologic analysis, Delphi survey [30], and technology roadmap [19], which is under the guidance of expert knowledge and experience and requires a lot of human participation. Therefore, qualitative methods have high research costs and subjectivity, making the research results inefficient and unstable. With the rapid growth of data mining technology, quantitative methods have been well applied in technology evolution analysis. The main quantitative analysis methods include patent citation analysis, patent classification analysis, text mining methods, etc.

Huenteler et al. analyzed the evolution process of technology based on the citation links of patents, while a citation network can reflect the flow process of knowledge [31]. Zhou et al. analyzed the technology layout and trends of solar cells based on patent classification by IPC code [32]. However, the citation network analysis and classification analysis do not take semantic information in the text corpus into consideration. Additionally, the IPC code does not change over time. Thus, it is unable to sensitively perceive the technology evolution for the rapidly developing or converging and emerging technology fields. To fully use the semantic information in patent text, text mining methods were taken into consideration to analyze technology evolution. Yoon et al. constructed a semantic network using text mining methods to analyze the development trend of technology [33]. Miao et al. has studied more than 30,000 patents since the 1990s using text mining methods to obtain products and applications with application prospects and rule out traditional technologies with a declining trend [11]. However, text mining methods pay more attention to the semantic information carried by patent text while ignoring the relationship between patents. Naturally, researchers consider combining the patent citation network and text mining methods to research technology evolution trends. Li et al. monitored and forecast the development trend of nanogenerators by citation analysis and used a Hierarchical Dirichlet Process topic model to extract technological topics [8].

Moreover, most of the existing studies only focus on a single source of data such as patents and papers, ignoring the interaction between knowledge discovery represented by grants or papers and technologies applications represented by patents, as well as the correction and difference of the information.

### 2.3. Knowledge Graph and Representation Learning

With the advent of the information age, the explosive growth of multi-source heterogeneous data has brought significant challenges to the data organization and application in the big data environment. A knowledge graph (KG) is a structured knowledge base with strong semantic processing ability, which provides a new idea to solve these problems. KG comes from Google’s next-generation intelligent semantic search engine technology. In essence, it is a semantic network that reveals the relationship between entities and can also formally describe things that existed in the real world and their relationships. Now KG has been used to refer to all kinds of large-scale knowledge bases. Within the KG, the storage structure of data and knowledge is a triple, such as <*s*, *p*, *o*> or *p* (*s*, *o*), where *s* and *o* are nodes in the KG, representing subject entity knowledge and object entity knowledge, respectively, and *p* is the edge in the KG, meaning the relational knowledge from subject *s* to object *o*.

At present, general knowledge graph technology, such as Freebase, DBpedia, Wikidata, and so on, has played an essential role in the internet field, such as intelligent search, intelligent Q&A, and personalized recommendation. At the same time, it has been preliminarily applied in many areas such as finance, e-commerce, medical treatment, etc. Compared with the general knowledge graph, the domain knowledge graph has more knowledge sources, faster requirements for large-scale expansion, a more complex knowledge structure, higher requirements for knowledge quality, and broader application forms. In the field of nanogenerators, there is little literature on the application of knowledge to analyze the relationships between various entities.

A knowledge graph is a structured knowledge base that stores entities’ features and relationships, which demands a data mining method to efficiently obtain specific knowledge from the vast knowledge base. In recent years, representation learning algorithms have developed rapidly. Their purpose is to learn the potential, informative, and low dimensional representation of entities, which can simplify the graph while retaining the graph structure, entities’ features, labels, and other auxiliary information. Socher et al. defined the evaluation function for each triplet in the knowledge graph using a single-layer neural network. They solved the representation of each entity by maximizing the evaluation function [34]. Although the nonlinear model based on the single-layer neural network can capture the semantic relationship between entities well, the computational cost is considerable. Inspired by the phenomenon of translation invariance in word vector space, Bordes et al. proposed the TransE model to learn the representation of entities in the knowledge graph in vector space, and the relationship is regarded as the translation vector between related entity pairs to constrain the learning results [35]. The TransE model is simple to reduce the computational cost, and the performance is significantly improved compared with the previous models. Nevertheless, TransE still has many limitations, which has encouraged later researchers to put forward many improved models. Wang et al. thought that the same entity should have different vector representations under different relationships, so they proposed the TransH model to improve the ability to deal with complex relationships [36]. Lin et al. further proposed the TransR model based on the belief that different relationships should correspond to different semantic spaces [37]. The TransR model represents entities in triples into the vector space corresponding to the relationships and then establishes the translation relationship between entity vectors proposed by the TransE model. On the basis of TransR, the TransD model further defines different projection matrices for head entity and tail entity and simplifies the number of parameters of matrix [38].

TransE and its improved model only use the relationship data between entities in the knowledge graph for representation and learning. However, a large amount of descriptive information about the entity itself has not been used. Occasionally, the graph neural network (GNN) has attracted the attention of relevant researchers. GNN is a deep learning model based on information propagation, which can use the structure information and node information of the graph for representation at the same time. However, most classical GNN models, such as GCN [39], GAT [40], GAE [41], etc., can only apply to the knowledge graph of a single type of entity and relationship. To deal with this, Cen et al. proposed the MEIRec model, which uses the meta-path sampling method to sample multiple subgraphs of unified formal structures to facilitate GNN representation learning [42]. Wang et al. proposed the HAN model, which calculates the adjacency matrix of different meta-paths and puts it into the GAT model to learn the graph representation [43].

## 3. Methods

### 3.1. Data

This study attempted to analyze the knowledge flow between different data sources. Firstly, using the term “nanogenerator* or nano-generator”, we collected the papers of nanogenerators in the Thomson Reuters Web Of Science database (WOS) by the end of December 2021. Then, 3304 publications were retrieved from the whole database, including the publication’s title, citation information, abstract, time, author, institution, DOI, and journal name. Likewise, using the term “nanogenerator* OR nanometer generator”, we collected the patents and nanogenerators in the Derwent Innovation Index (DI) database by the end of December 2021. Then, 984 patents were retrieved from the database, including the patent’s title, citation information, time, and institution. Finally, using the term “nanogenerator*”, we collected the grants of nanogenerators in the grants database of the China Knowledge Centre for Engineering Science and Technology (CKCEST). A total of 169 grants were retrieved, including title, start date, keywords, abstract, and institution. The details of data acquisition are shown in Table 1.

### 3.2. Knowledge Graph of Different Time Slices

To make use of the semantic information in the multi-source data, we need to construct knowledge graphs to reflect the relationships between entities. Take paper data as an example. Based on the related entities of papers, such as author, institution, and journal, we can construct a mapping *r* (*s*, *o*) to preserve the relationship of paper and other entities, while *s* represents the source of the relationship and *o* represents the object of the relationship, and *r* represents the type of relationship. Then, we can obtain several relationships, such as papers published in a journal *p* (*p*, *j*), papers written by the author *w* (*p*, *a*), and papers owned by an institution *o* (*p*, *i*). In the meantime, by dealing with the citation information of papers, we can obtain the relationship of a paper cited by other papers *c* (*p*, *p*). For each type of relationship, we can construct a matrix $M_{AB}$ to save the mapping, while *A* and *B* represent the type of entities.

Thus far, we have obtained the relationships between entities by the semantic information contained in papers. Next, we need to extract features that can reflect the similarity and differences of papers by the word vectorization method. Specifically, we can vectorize the title of papers by the doc2vec model (denoted by $f_{i}$). After vectorization, the paper with similar subject words in the title has higher vector similarity, which saves the feature information of papers. The process of knowledge graph construction of patents and grants is the same as that of papers.

After constructing the knowledge graph of different data sources, we cut it into three time-slices of 2006–2012, 2013–2017, and 2018–2021. The detail of the knowledge graph is shown in Table 2. While 2006–2012 represents the preliminary stage of nanogenerators because PENG was proposed in 2006 and TENG was proposed in 2012, 2013–2017 represents the development stage of nanogenerators, and 2018–2021 represents the present stage.

### 3.3. Heterogeneous Graph Attention Network for Representation Learning

In this paper, we use a Heterogeneous Graph Attention Network (HAN) to consider the graph topology and text information at the same time [43]. The HAN model is improved from the Graph Attention Network (GAT) model while reserving the attention mechanism of GAT and proposing a solution for heterogeneous graph representation learning [40]. The framework of HAN is shown in Figure 2.

First, the meta-path was defined as a path in the form of $E_{1}\;\overset{R_{1}}{\to }\;E_{2}\;\overset{R_{2}}{\to }\cdots \;\overset{R_{n}}{\to }\;E_{n+1}$ (abbreviated as $E_{1}E_{2}\cdots E_{n+1}$), which describes the composite relation $R=R_{1}\;{}^{\circ}\;R_{2}\;{}^{\circ}\cdots \;{}^{\circ}\;R_{n}$ between entities $E_{1\;}$ and $E_{n+1}$, where ° denotes the composition operator on relations.

Based on the definition of meta-path, we can extract relations between different papers, grants, or patents. For example, we can define the relation of journal co-occurrence of papers by the meta-path $P_{1}\;\overset{published}{\to }\;J_{1}\;\overset{publish}{\to }\;P_{2}$ (abbreviated as *PJP*). The complete meta-paths of different data sources are shown in Table 2. Specifically, based on the relationship we obtained in the process of knowledge graph construction, we can calculate the transformation matrix of different meta-paths by matrix multiplication ($M_{PP}=M_{PJ}\times M_{JP}$, $M_{PP}$ can be denoted by $M_{\varphi i}\;$ while φi represent the type of entities).

Next, based on the transformation matrix of different meta-paths, for each type of entity (e.g., entities with type φi), we can conduct information propagation process as follows:(1)$$f_{i}^{\prime }=M_{\varphi i}\;\cdot \;f_{i}$$
where $f_{i}$ and $f_{i}^{\prime }$ are the original and processed features of node *i*, respectively.

After that, self-attention is leveraged to learn the weight among various kinds of entities. Given an entity pair (*i*, *j*) which are connected via meta-path φ, a node-level attention $\alpha_{ij}^{\varphi }$ can be learned to show how important entity *j* will be for entity *i*. The process can be formulated as follows:(2)$$\alpha_{ij}^{\varphi }=att_{node}\left(f_{i}^{\prime },\;f_{j}^{\prime },\;\varphi \right)$$

Then, the meta-path-based embedding of entity *i* can be aggregated by the neighbor’s projected features with the corresponding coefficients as follows:(3)$$z_{i}^{\varphi }=\sigma \left(\underset{j\epsilon N_{i}^{\varphi }}{\sum }\alpha_{ij}^{\varphi }\;\cdot \;f_{j}^{\prime }\right)$$
where $z_{i}^{\varphi }$ is the learned embedding of entity *i* for meta-path φ.

Given the meta-path set {$\varphi_{1},\varphi_{2},\cdots \varphi_{m}$}, after feeding features into entity-level attention, we can obtain m groups of semantic specific node embeddings, denoted as {$Z_{\varphi 1},Z_{\varphi 2},\cdots Z_{\varphi m}$}.

Generally, every node contains multiple types of semantic information, and semantic entity embedding can only reflect nodes from one aspect. To learn a more comprehensive node embedding, we need to fuse multiple semantics, which can be revealed by meta-paths. A novel semantic-level attention was proposed to automatically learn the importance of different meta-paths and fuse them. The learned weights of each meta-path can be shown as follows:(4)$$\left(\beta_{\varphi 1},\;\beta_{\varphi 2},\cdots \beta_{\varphi m}\right)=att_{sem}\left(Z_{\varphi 1},\;Z_{\varphi 2},\cdots ,Z_{\varphi m}\right)$$

With the learned weights as coefficients, we can fuse these semantic-specific embeddings to obtain the final embedding *Z* as follows:(5)$$Z=\overset{M}{\underset{m=1}{\sum }}\beta_{\varphi m}\;\cdot \;Z_{\varphi m}$$

### 3.4. K-Means for Clustering and LDA for Topic Extracting

K-means is an unsupervised clustering algorithm, which identifies clusters $C=\left\{C_{1},\;C_{2},\cdots ,C_{k}\right\}$ based on square error minimization for the given sample set $D=\left\{x_{1},x_{2},\cdots ,x_{n}\right\}$. The process can be expressed as:(6)$$E=\overset{k}{\underset{i=1}{\sum }}\underset{x\;\epsilon C_{i}}{\sum }||x-\mu_{i}||{}^{2}$$
where $\mu_{i}=\frac{1}{\left|C_{i}\right|}\underset{x\epsilon C_{i}}{\sum }x$ is the mean vector for cluster $C_{i}$, and *k* is the number of clusters proposed to be classified.

In this paper, the final embedding of entities was used as the input of the K-means model for clustering. Then, we can obtain k clusters, which represent research sub-fields.

To clarify what each cluster means, we used the Latent Dirichlet Distribution (LDA) topic model to extract topic words for clusters. The LDA topic model is an unsupervised method for extracting hidden topics distribution of document and hidden word distribution of topics. It can represent each cluster by several important topics, and each topic contains several keywords.

### 3.5. Clusters Association for Evolutionary Path Identification

The mean value of entity embedding vectors was calculated to reflect the cluster vector. By calculating the similarity of different cluster vectors in different time slices or different data sources, we can connect clusters with the highest similarity to form technology evolution paths, in which the clusters’ topics were used to reflect specific technologies. In this paper, the reciprocal of the Euclidean distance was used to measure the similarity of different clusters.

## 4. Results and Discussions

### 4.1. Representation Learning and Clustering

According to the proposed method in Section 3, the technology evolution pathway was identified and described. The multi-source data were utilized to construct the knowledge graphs of different data sources and different time slices. Based on these knowledge graphs, we can extract the transformation matrix A∈ℝn×n by different meta-paths, and the feature matrix X∈ℝn×m by doc2vec model, while n was the number of grant, paper, or patent entities in the knowledge graph, which can be found in Table 2 and m was the vector dimension of doc2vec output.

Then, the transformation matrix *A* and feature matrix *X* were input into the HAN model to learn the representation vector of entities. In this paper, we set the learning rate to 0.005, the dimension of the semantic-level attention vector to 128, the attention head K to 8, the dropout of attention to 0.6, and the training epochs to 200.

After using the trained model to get embedding vectors with 64 dimensions, we utilized K-means model to cluster these embedding vectors. In order to select the number of clusters accurately, we chose the number corresponding to the maximum silhouette coefficient while repeating clustering for cluster number change in ranges 2 to 10.

After clustering, we extracted keywords of clusters by LDA topic model using the text information in each cluster. We provide one topic and ten keywords for each cluster. The details can be found in Table 3, Table 4 and Table 5.

All of the experimental procedures were based on Python 3 programming language and PyCharm platform.

From Table 3, Table 4 and Table 5, we can summarize the technology topic of different time slices. In 2006–2012, the main topic was the PENG structure and sensors based on PENG. In 2013–2017, the flexible sensors and wearable devices were the mainstream nanogenerator applications, while TENG began to appear and gradually replace PENG. In 2018–2021, wearable devices were still the research hotspots, while novel energy sources and the performance improvement of nanogenerators such as output voltage became the research questions.

### 4.2. Technology Evolution Pathways

Following the step of K-means, we calculate the vector distance of clusters in different time slices and connect the clusters with minimum distance while the minimum distance is smaller than the threshold (set to 2) to form the technology evolution pathways. The results are shown in Figure 3, in which the evolution pathways were automatically generated by calculating the similarity of the preceding clustering results using a written Python program. The dots in Figure 3 indicate the clusters which connect similar grants, papers, and patents. The line connections between dots indicate high similarity between different clusters, which can represent the knowledge flow and indicate the technology evolution pathways.

From Figure 3, we can analyze the knowledge flow pathways between data sources. First, we can find that the knowledge flows from grants to papers were faster than that from papers to patents, as the technologies proposed by grants can be found in papers in the same time slice but can be found in patents in the backward time slice. An explanation for this condition is that making a profound study is easier than applying theory to application.

Then, we can find several knowledge flows from research to application successfully. The most typical case is the wearable devices with nanogenerator sensors. Wearable devices were proposed by grants in cluster 2 in 2013–2017 based on the basic research of nanogenerator structures and materials, and then get a profound study by scholars in 2013–2017. Finally, after abundant research about the performance of flexible sensors and the development of remote monitoring and communication technology, wearable devices based on flexible and self-powered nanogenerators were applied in daily life. In addition, based on Figure 3, we can also monitor the evolutionary pathways of piezoelectric nanogenerators.

Except for these obvious evolution pathways, we can find several isolated short pathways in 2018–2021, which represent the technologies with strong innovativeness. Specifically, cluster 3 of papers in 2018–2021 contains the keywords “mechanical”, “wave”, “wind”, “water”, “vibration”, “energy” and “harvesting”, and cluster 4 of patents in 2018–2021 contains the keywords “water” and “energy”. These keywords demonstrate that novel energy sources such as wind, water, and mechanical vibration became new research directions and hotspots. Cluster 0 of papers in 2018–2021 represents the fiber structure of nanogenerators, which indicates the innovation direction of nanogenerator structures.

## 5. Conclusions

This paper proposed a novel framework to monitor the evolutionary pathways of nanogenerator technology based on multi-source data and a knowledge graph. In the framework, the knowledge graph makes full use of text information, and the multi-source data fully considers the evolutionary pathways from different data perspectives. Additionally, we show that the novel framework is efficient and accurate.

We find some characteristics that the evolution process and knowledge flow from grants to patents is faster than that from papers to patents, which indicates that making a profound study is easier than applying theories to applications. We also monitor the complete evolution pathways of piezoelectric nanogenerators, wearable devices, and nanogenerator performance improvement technologies. While analyzing the evolution pathways, we also find several emerging research directions for nanogenerators, such as novel energy sources and fiber structure of nanogenerators.

However, due to the numbers of grants, papers, and patents in the nanogenerator field, we cannot unleash the full advantage of the knowledge graph and representation learning. In the meantime, the identification of cluster topics requires expert knowledge and human intervention. So, in future research, we will attempt to get more data and use the machine learning method to achieve the automatic classification of cluster topics.

## Figures and Tables

**Figure 1 nanomaterials-12-00838-f001:**
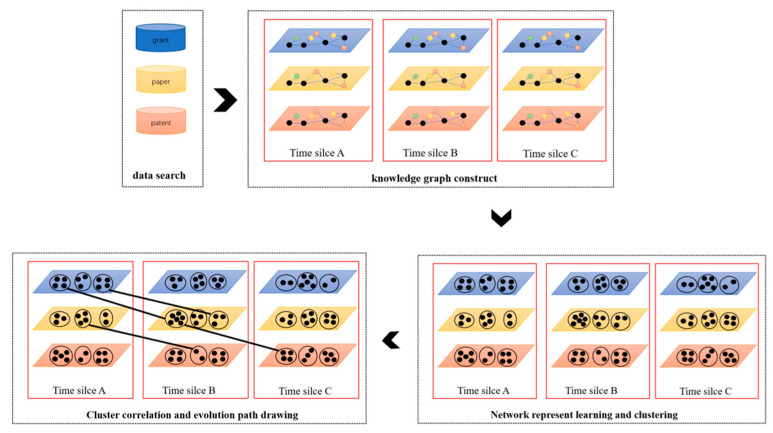
The framework to monitor the technology evolution pathways.

**Figure 2 nanomaterials-12-00838-f002:**
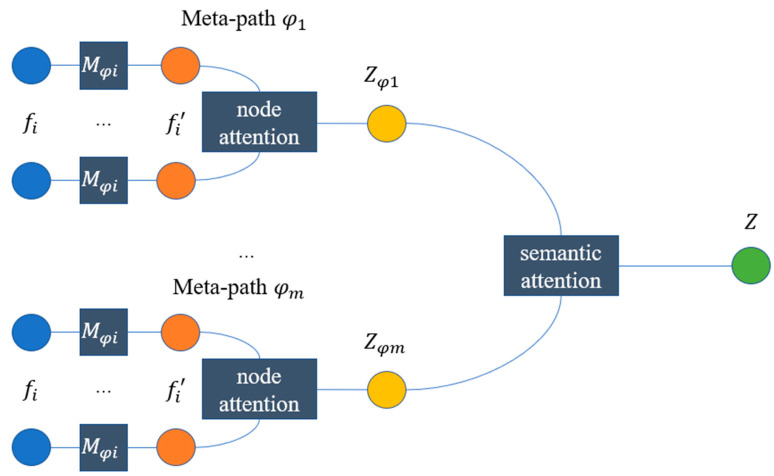
The framework of HAN.

**Figure 3 nanomaterials-12-00838-f003:**
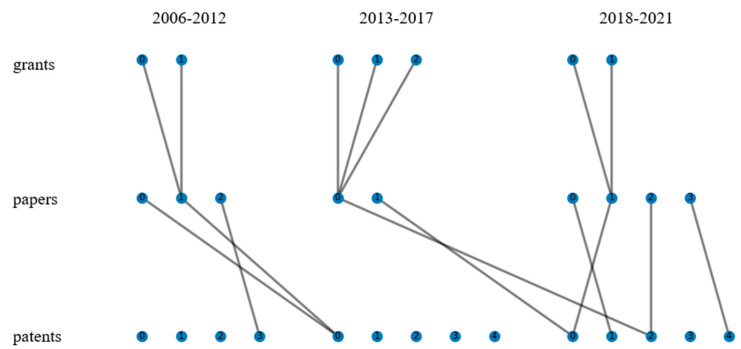
Evolution pathways between different data sources.

**Table 1 nanomaterials-12-00838-t001:** Description of data acquisition.

Data	Database	Time Range	Search Query	Amounts
Grants	China Knowledge Centre for Engineering Science and Technology (CKCEST)	2006–2021	nanogenerator*	169
Papers	Thomson Reuters Web Of science database (WOS)	2006–2021	TI = (nanogenerator* OR nano-generator*) AND PY = (2006–2021)	984
Patents	Derwent Innovation Index database (DI)	2006–2021	TI = (nanogenerator* OR nanometer generator) AND PY ≤ 2021	3304

**Table 2 nanomaterials-12-00838-t002:** Description of knowledge graph and meta-path selection.

Data Source	Time Slice	Number of Entities	Types of Relations	Meta-Paths
Grants	2006–2012	21	Contain (grant, keyword) Own (institution, grant)	G-K-G G-I-G
	2012–2017	76		
	2017–2021	33		
Papers	2006–2012	134	Publish (journal, paper) Write (author, paper) Cite (paper, paper) Own (institution, paper)	P-J-P P-A-P P-I-P P-P
	2013–2017	825		
	2017–2021	2345		
Patents	2006–2012	105	Cite (patent, patent) Own (institution, patent)	P-P P-I-P
	2013–2017	337		
	2017–2021	542		

**Table 3 nanomaterials-12-00838-t003:** Cluster information of grants.

Data Source	Time Slice	Cluster Number	Numbers of Entities	Keywords	Categories
Grants	2006–2012	0	19	nanometer, nanogenerator, structure, development, characteristic, application, utilize, piezoelectric, analysis, nanowire	PENG structure
Grants	2006–2012	1	2	nanometer, influence, wide band gap, energy, structure, research, characteristic, photoelectricity, stress, element	Undefined
Grants	2013–2017	0	55	nanogenerator, friction, drive, sensor, flexible, nanomaterial, structure, electric, piezoelectric, biology	PENG applications
Grants	2013–2017	1	12	piezoelectric, nanogenerator, ZnO, element, energy, structure, harvest, nanowire, power supply	PENG structure
Grants	2013–2017	2	9	nanometer, friction, structure, regulation, semiconductor, device, polymer, wearable, nanomaterial	Wearable devices
Grants	2018–2021	0	12	nanogenerator, structure, piezoelectric, wearable, biology, power supply, nanometer, element, application, detection	PENG applications
Grants	2018–2021	1	21	nanometer, friction, research, nanogenerator, harvest, performance, energy, mechanism, flexibility, application	TENG applications

**Table 4 nanomaterials-12-00838-t004:** Cluster information of papers.

Data Source	Time Slice	Cluster Number	Numbers of Entities	Keywords	Categories
Paper	2006–2012	0	94	nanogenerator, piezoelectric, ZnO, flexible, transparent, sensor, nanowire, self-powered, array, substrate	PENG applications
Paper	2006–2012	1	28	nanogenerator, piezoelectric, nanostructure, ZnO, ultrasound, piezotronics, energy, nano-systems, oxide, self-powered	PENG structure
Paper	2006–2012	2	12	nanogenerator, self-powered, piezoelectric, graphene, alpha-particle, driven, actinium255, sensor, ZnO, energy	PENG materials
Paper	2013–2017	0	454	nanogenerator, triboelectric, energy, self-powered, harvesting, piezoelectric, sensor, flexible, wearable, system	Wearable devices
Paper	2013–2017	1	371	nanogenerator, triboelectric, piezoelectric, flexible, based, output, performance, effect, enhanced, application	Performance improvement
Paper	2018–2021	0	390	nanogenerator, piezoelectric, triboelectric, energy, harvesting, performance, composite, electrospun, nanofibers	Fiber structure
Paper	2018–2021	1	530	triboelectric, nanogenerator, performance, high, output, effect, charge, enhanced, effect, density	Performance improvement
Paper	2018–2021	2	918	triboelectric, nanogenerator, self-powered, sensor, wearable, flexible, system, monitoring, stretchable, motion	Wearable devices
Paper	2018–2021	3	507	triboelectric, nanogenerator, energy, harvesting, self-powered, mechanical, wave, water, wind, vibration	Energy source

**Table 5 nanomaterials-12-00838-t005:** Cluster information of patents.

Data Source	Time Slice	Cluster Number	Numbers of Entities	Keywords	Categories
Patent	2006–2012	0	9	bubble, generator, treatment, water, method, involves, utilizing, based, micro-nano, controlled, nano	undefined
Patent	2006–2012	1	18	layer, zinc, substrate, piezoelectric, oxide, element, laminating, manufacturing, method, nanowire	Manufacturing method of PENG layers
Patent	2006–2012	2	35	piezoelectric, nanogenerator, structure, solar, power, electrical, conductive, energy, material, cell	PENG structure
Patent	2006–2012	3	43	electrode, layer, nanogenerator, substrate, piezoelectric, array, insulating, material, power, film	PENG materials
Patent	2013–2017	0	58	nanogenerator, energy, piezoelectric, element, storing, comprises, layer, substrate, electric, storage	PENGstructure
Patent	2013–2017	1	99	friction, layer, electrode, generator, nanogenerator, power, component, nano, surface, signal	TENG structure
Patent	2013–2017	2	56	friction, layer, triboelectric, nanogenerator, electrode, conductive, unit, power, generator, surface	TENG structure
Patent	2013–2017	3	60	layer, nanogenerator friction, electrode, film, polymer, piezoelectric, material, metal, flexible	TENG materials
Patent	2013–2017	4	64	generator, friction, device, energy, flexible, power, electric, nanogenerator, storage, nanometer	TENG application
Patent	2018–2021	0	128	friction, nanogenerator, connected, signal, electrode, system, layer, sensor, voltage, output	Performance improvement
Patent	2018–2021	1	106	triboelectric, nanogenerator, layer, film, piezoelectric, composite, material, electrode, flexible, generator	TENG materials
Patent	2018–2021	2	118	friction, nanogenerator, layer, energy, wearable, device, conductive, triboelectric, body, power	Wearable devices
Patent	2018–2021	3	87	layer, friction, electrode, nanogenerator, substrate, conductive, flexible, structure, material, comprises	TENG structure
Patent	2018–2021	4	103	friction, nanogenerator, generator, device, energy, water, plate, shaft, inner, connected, layer	TENG energy harvesting

## Data Availability

Not applicable.

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
