# Peer review of "Advanced Technology Evolution Pathways of Nanogenerators: A Novel Framework Based on Multi-Source Data and Knowledge Graph"

_nanomaterials, 2022, doi:10.3390/nano12050838_

Round 1
Reviewer 1 Report
Authors present interesting correlation between grants, papers and patents in the field of nanogenerators. The manuscript can be published after minor revisions:
- There area some misprints, that should be corrected. The captures of the figures should begin by capital letter.
- Sentence: “Notably, nanogenerators present widespread applications other than energy harvesting, benefiting from related technologies such as 5G, Internet of Things (IoT), nanomaterials (DOI: 10.3390/POLYM12122766), flexible sensors, and so on.” Please, provide literaturee for every technology that you mentioned, I suggested only for one of them.
- I recommend to present statistical data about amount of papers and patents in form of chart. It is easies for perception.
- Please, add to the test what key-words did you used for making the search at the mentioned database (Thomson Reuters Web Of science database) or other databases. Also add the kind of logical operator between these words.
Author Response
First of all, we sincerely thank you for reviewing our paper and providing these valuable comments and hints for improving its contents and presentation. In accordance with the constructive comments and suggestions, we have revised our manuscript seriously. In the following, we detail how the paper has been adapted and modified in red color, including detailed answers to all the comments received.
Reviewer #1: Authors present interesting correlation between grants, papers and patents in the field of nanogenerators. The manuscript can be published after minor revisions:
There are some misprints, that should be corrected. The captures of the figures should begin by capital letter.
Response: Thank you for the comment. We have replaced some captures of figures and tables with capital letters.
Sentence: “Notably, nanogenerators present widespread applications other than energy harvesting, benefiting from related technologies such as 5G, Internet of Things (IoT), nanomaterials (DOI: 10.3390/POLYM12122766), flexible sensors, and so on.” Please, provide literature for every technology that you mentioned, I suggested only for one of them.
Response: Thank you for the comment. We have provided literature for every technology. And we have merged 5G and IoT because they are associated.
Relevant literature can be found in the manuscript (cite number “3”, “4”, “5”):
5G and Internet of Things:
Zhao, X.; Askari, H.; Chen, J. Nanogenerators for smart cities in the era of 5G and Internet of Things. Joule 2021, 5, 1391-1431, doi: 10.1016/j.joule.2021.03.013.
Nanomaterials:
Mahapatra, B.; Kumar Patel, K.; Vidya; Patel, P.K. A review on recent advancement in materials for piezoelectric/triboelectric nanogenerators. Materials Today: Proceedings 2021, 46, 5523-5529, doi: 10.1016/j.matpr.2020.09.261.
Flexible sensors:
Zhang, D.; Xu, Z.; Yang, Z.; Song, X. High-performance flexible self-powered tin disulfide nanoflowers/reduced graphene oxide nanohybrid-based humidity sensor driven by triboelectric nanogenerator. Nano Energy 2020, 67, doi: 10.1016/j.nanoen.2019.104251.
I recommend to present statistical data about amount of papers and patents in form of chart. It is easies for perception.
Please, add to the test what key-words did you used for making the search at the mentioned database (Thomson Reuters Web Of science database) or other databases. Also add the kind of logical operator between these words.
Response: Thank you for the comment. We have added a chart to show the amount of grants, papers, and patents (shown in Table 1). And in the text and chart, we have added the search query while searching at the mentioned database.

Reviewer 2 Report
The author done very good work to understand the era of nanogenerators and proposed a novel framework to monitor the evolutionary pathways of nanogenerator technology based on multi-source data and knowledge graphs. But the author should consider the following and include them in the article for better understanding.
Review Comments:
⦁ The author should include the references referred for the novel framework to monitor the technology evolution pathways which was shown in figure 1.
⦁ What does the color and the black dots indicate in figure 1 and need an explanation about figure 1?
⦁ The author can design a pie chart for a better understanding of figure 2.
⦁ The author please cite the reference for the HAN network and GAT model.
⦁ In table 2, the author should change the name of the last column from “topic” to some other names like categories, etc., because it seems like a key hint of the topic.
⦁ Does the author used any software to design the figure 3, and what does the dots and line connections between grants, papers, and patents indicates?
⦁ The author please explain the sentence in section 4.2 “Specifically, as cluster 3 of papers and cluster 4 of patents in 2018-2021 represents, novel energy sources like wind, water, and mechanical vibration became a new research direction and hotspots.”
⦁ The authors should cite a few papers related to this work.
https://www.sciencedirect.com/science/article/pii/S2211285519307232
Author Response
First of all, we sincerely thank you for reviewing our paper and providing these valuable comments and hints for improving its contents and presentation. In accordance with the constructive comments and suggestions, we have revised our manuscript seriously. In the following, we detail how the paper has been adapted and modified in red color, including detailed answers to all the comments received.
Reviewer #2: The author done very good work to understand the era of nanogenerators and proposed a novel framework to monitor the evolutionary pathways of nanogenerator technology based on multi-source data and knowledge graphs. But the author should consider the following and include them in the article for better understanding.
Review Comments:
⦁ The author should include the references referred for the novel framework to monitor the technology evolution pathways which was shown in figure 1.
Response: Thank you for the comment. We have added reference referred to the novel framework:
Miao, Z.; Du, J.; Dong, F.; Liu, Y.; Wang, X. Identifying technology evolution pathways using topic variation detection based on patent data: A case study of 3D printing. Futures 2020, 118, doi: 10.1016/j.futures.2020.102530.
Liu, H.; Chen, Z.; Tang, J.; Zhou, Y.; Liu, S. Mapping the technology evolution path: a novel model for dynamic topic detection and tracking. Scim 2020, 125, 2043-2090, doi:10.1007/s11192-020-03700-5.
Zhou, Y.; Dong, F.; Kong, D.; Liu, Y. Unfolding the convergence process of scientific knowledge for the early identification of emerging technologies. Technol. Forecast. Soc. Change 2019, 144, 205-220, doi: 10.1016/j.techfore.2019.03.014.
Kong, D.; Yang, J.; Li, L. Early identification of technological convergence in numerical control machine tool: a deep learning approach. Scim 2020, 125, 1983-2009, doi:10.1007/s11192-020-03696-y.
You can find it in cite number “11”, “12”, “13”, “14”.
⦁ What does the color and the black dots indicate in figure 1 and need an explanation about figure 1?
Response: Thank you for the comment. The background colors represent different data sources. The different colors of dots represent different types of entities in the knowledge graph (like author, paper, institution, journal in paper knowledge graph), and the connections between dots represent the relations between different entities.
The explanation about figure 1 is shown below:
After multi-source data acquisition, the knowledge graph was constructed to capture semantic information between entities as shown in the top right corner of Figure 1, while different colors of dots show the different types of entities (like author, paper, institution, and journal in paper knowledge graph), and the connections between dots show the relations between entities. Then representation learning and clustering methods were used to cluster entities with similar topics as shown in the bottom right corner in Figure 1, while the circles represent clusters and the black dots represent the grants, papers, and patents contained in clusters. Finally, we described the evolutionary path from grants to papers and then to patents by connecting similar clusters.
⦁ The author please cite the reference for the HAN network and GAT model.
Response: Thank you for the comment. The HAN and GAT have been cited in section 2.3 while doing literature view (cite numbers “38” and “35”). And we further added the cited number in section 3.3.
⦁ In table 2, the author should change the name of the last column from “topic” to some other names like categories, etc., because it seems like a key hint of the topic.
Response: Thank you for the comment. We have changed the column name “topic” to “categories”.
⦁ Does the author used any software to design the figure 3, and what does the dots and line connections between grants, papers, and patents indicates?
Response: Thank you for the comment. The whole experimental procedures were based on Python 3 programming language and PyCharm platform, including the data preprocessing, knowledge graph construct, representation learning, clustering, and the drawing of evolution pathways. Figure 3 was automatically designed based on preceding clustering results by the written Python program using “matplotlib.pyplot” package in Python.
The dots in figure 3 indicate the clusters which connect similar grants, papers, and patents to reflect specific technology topics. The line connections between dots indicate high similarity between different clusters, which can represent the knowledge flows and indicate the technology evolution pathways.
⦁ The author please explain the sentence in section 4.2 “Specifically, as cluster 3 of papers and cluster 4 of patents in 2018-2021 represents, novel energy sources like wind, water, and mechanical vibration became a new research direction and hotspots.”
Response: Thank you for the comment. The first cluster mentioned above can be found in the last row of Table 4 and the second cluster can be found in the last row of Table 5. In these clusters, the keywords of papers contain “mechanical”, “wave”, “wind”, “water”, “vibration”, “energy”, “harvesting” and the keywords of patents contain “water”, “energy”, which representing the research of novel energy sources. And the papers and patents in these clusters are published in 2018-2021, which can reflect the research direction and hotspots at present.
⦁ The authors should cite a few papers related to this work.
https://www.sciencedirect.com/science/article/pii/S2211285519307232
Response: Thank you for the comment. We have cited the paper you recommended. You can find it in cite number “21”.
